OBSERVATION

# Inflammatory Endotype-Associated Airway Resistome in Chronic Obstructive Pulmonary Disease

Xinzhu Yi,[a] Yanjun Li,[b] Haiyue Liu,[c] Xiaomin Liu,[a] Junhao Yang,[a] Jingyuan Gao,[a] Yuqiong Yang,[d] Zhenyu Liang,[d] Fengyan Wang,[d] Dandan Chen,[e] Lingwei Wang,[e] Weijuan Shi,[d] David C. L. Lam,[f] Martin R. Stampfli,[g] Paul W. Jones,[h] Rongchang Chen,[d,e] ⓘ Zhang Wang[a]

[a]Institute of Ecological Sciences, School of Life Sciences, South China Normal University, Guangzhou, China
[b]Emergency Department, Shenzhen Traditional Chinese Medicine Hospital, Shenzhen, China
[c]Xiamen Key Laboratory of Genetic Testing, Department of Laboratory Medicine, the First Affiliated Hospital of Xiamen University, Xiamen, China
[d]First Affiliated Hospital of Guangzhou Medical University, National Clinical Research Center for Respiratory Disease, Guangzhou Institute of Respiratory Health, State Key Laboratory of Respiratory Disease, Guangzhou, China
[e]Key Laboratory of Shenzhen Respiratory Diseases, Institute of Shenzhen Respiratory Diseases, Shenzhen People's Hospital, Shenzhen, China
[f]Department of Medicine, Li Ka Shing Faculty of Medicine, University of Hong Kong, Hong Kong SAR, China
[g]Department of Medicine, Firestone Institute of Respiratory Health at St. Joseph's Healthcare, McMaster University, Hamilton, Ontario, Canada
[h]Institute for Infection and Immunity, St George's, University of London, London, United Kingdom

Xinzhu Yi, Yanjun Li, and Haiyue Liu contributed equally to this article. Author order was determined by the type of their contributions.

**ABSTRACT** Antimicrobial resistance is a global concern in chronic respiratory diseases, including chronic obstructive pulmonary disease (COPD). The collection of antibiotic resistance genes or resistome in human airways may underlie the resistance. COPD is heterogeneous, and understanding the airway resistome in relation to patient phenotype and endotype may inform precision antibiotic therapy. Here, we characterized the airway resistome for 94 COPD participants at stable disease. Among all demographic and clinical factors, patient inflammatory endotype was associated with the airway resistome. There were distinct resistome profiles between patients with neutrophilic or eosinophilic inflammation, two primary inflammatory endotypes in COPD. For neutrophil-predominant COPD, the resistome was dominated by multidrug resistance genes. For eosinophil-predominant COPD, the resistome was diverse, with an increased portion of patients showing a macrolide-high resistome. The differential antimicrobial resistance pattern was validated by sputum culture and *in vitro* antimicrobial susceptibility testing. *Ralstonia* and *Pseudomonas* were the top contributors to the neutrophil-associated resistome, whereas *Campylobacter* and *Aggregatibacter* contributed most to the eosinophil-associated resistome. Multiomic analyses revealed specific host pathways and inflammatory mediators associated with the resistome. The arachidonic acid metabolic pathway and matrix metallopeptidase 8 (MMP-8) exhibited the strongest associations with the neutrophil-associated resistome, whereas the eosinophil chemotaxis pathway and interleukin-13 (IL-13) showed the greatest associations with the eosinophil-associated resistome. These results highlight a previously unrecognized link between inflammation and the airway resistome and suggest the need for considering patient inflammatory subtype in decision-making about antibiotic use in COPD and broader chronic respiratory diseases.

**IMPORTANCE** Antibiotics are commonly prescribed for both acute and long-term prophylactic treatment in chronic airway disorders, such as chronic obstructive pulmonary disease (COPD), and the rapid growth of antibiotic resistance is alarming globally. The airway harbors a diverse collection of microorganisms known as microbiota, which serve as a reservoir for antibiotic resistance genes or the resistome. A comprehensive understanding of the airway resistome in relation to patient clinical and biological factors may help inform decisions to select appropriate antibiotics for clinical therapies.

**Ad Hoc Peer Reviewer** ⓘ Jianmin Chai, University of Arkansas

Address correspondence to Rongchang Chen, chenrc@vip.163.com, or Zhang Wang, wangz@m.scnu.edu.cn.

The authors declare a conflict of interest. X.Y., Y.L., H.L., X.L., J.Y., J.G., Y.Y., Z.L., F.W., D.C., L.W., W.S., R.C., and Z.W. have nothing to declare. M.R.S. is an employee of CSL Behring. P.W.J. is an employee of GlaxoSmithKline.

By deep multiomic profiling and *in vitro* phenotypic testing, we showed that inflammatory endotype, the underlying pattern of airway inflammation, was most strongly associated with the airway resistome in COPD patients. There were distinct resistome profiles between neutrophil-predominant and eosinophil-predominant COPD that were associated with different bacterial species, host pathways, and inflammatory markers, highlighting the need of considering patient inflammatory status in COPD antibiotic management.

**KEYWORDS** airway microbiome, antibiotic resistance, COPD, inflammation, macrolides

Chronic obstructive pulmonary disease (COPD) is one of the most prevalent respiratory diseases and is characterized by impaired lung function and persistent airway inflammation (1). COPD is heterogeneous, and there is a growing consensus for COPD management based on patient phenotype, endotype, and treatable traits (2, 3). Neutrophilic and eosinophilic inflammation are two major inflammatory endotypes with distinct pathogenesis, airway microbiota, and therapeutic options (4). Antimicrobial resistance is a concern in COPD since antibiotics are required for acute and long-term prophylactic therapy (5, 6). The COPD airway harbors a diverse microbiota that serve as a reservoir for antimicrobial resistance genes (ARGs) known as the resistome. As the airway resistome may underlie antibiotic resistance, understanding the resistome in relation to disease phenotype may help inform precision antibiotic therapy. Despite emerging efforts to unravel the airway resistome in chronic respiratory diseases (7, 8), it remains unclear how the resistome is related to clinical characteristics and host pathophysiology, limiting its applicability to clinical practice. One study has characterized the COPD resistome using quantitative PCR and reported a limited association between ARGs and clinical parameters (8). We hypothesize that the COPD airway resistome is inherently shaped by airway microbiota and is associated with patient inflammatory endotype. We report the use of deep metagenomic sequencing in conjunction with host multiomic characterization to assess the clinical and biological factors associated with the airway resistome in COPD.

Induced sputum from 99 stable COPD participants was collected in the First Affiliated Hospital of Guangzhou Medical University ($n = 72$) and Shenzhen People's Hospital ($n = 27$) in China (ethics approval reference numbers 2017-22 and KY-LL-2020294-01) and subjected to deep metagenomic sequencing (targeted >30 Gb of data per sample) with reagent controls (9). Five participants with missing records of long-term antibiotic use information were excluded from the analyses. All remaining 94 participants were free of antibiotics over the preceding 1 month, and none had long-term macrolide use. Participants were classified into neutrophilic, eosinophilic, and mixed- and pauci-granulocytic endotypes based on whether sputum neutrophil and eosinophil percentages were above or below established criteria (77.7% for neutrophil and 3% for eosinophil; Table S1 in the supplemental material) (10). A total of 307 ARGs across 18 antibiotic classes were obtained from metagenomic reads using ARGs-OAP (v2.0) (11) after quality filtering and host read removal using Sunbeam (supplemental methods) (12). Unsupervised clustering revealed four subtypes of the resistome, each with a distinct dominant antibiotic class ($\beta$-lactam-high, tetracycline-high, macrolide-lincosamide-streptogramin [MLS]-high and multidrug-high; Fig. 1a). A significant association was found between the resistome and microbiota composition using Procrustes analysis based on Bray-Curtis dissimilarity indices ($M^2 = 0.19$, $P < 0.001$; Fig. S1). Among all demographic and clinical features, the inflammatory endotype had a statistically significant, albeit modest, association with the resistome (Adonis, $R^2 = 0.087$, $P = 0.043$; Fig. 1b and c and Table S2). There was a reduced ARG diversity for participants in the neutrophilic subtype, whereas the total ARG abundance was higher for participants in the neutrophilic and mixed-granulocytic subtypes (Fig. 1d). The majority of participants (62.5%) in the neutrophilic subtype possessed a multidrug-high resistome, whereas participants in the eosinophilic subtype had a higher likelihood of harboring an MLS-high resistome (Fig. 1d). Despite a relatively greater proportion of

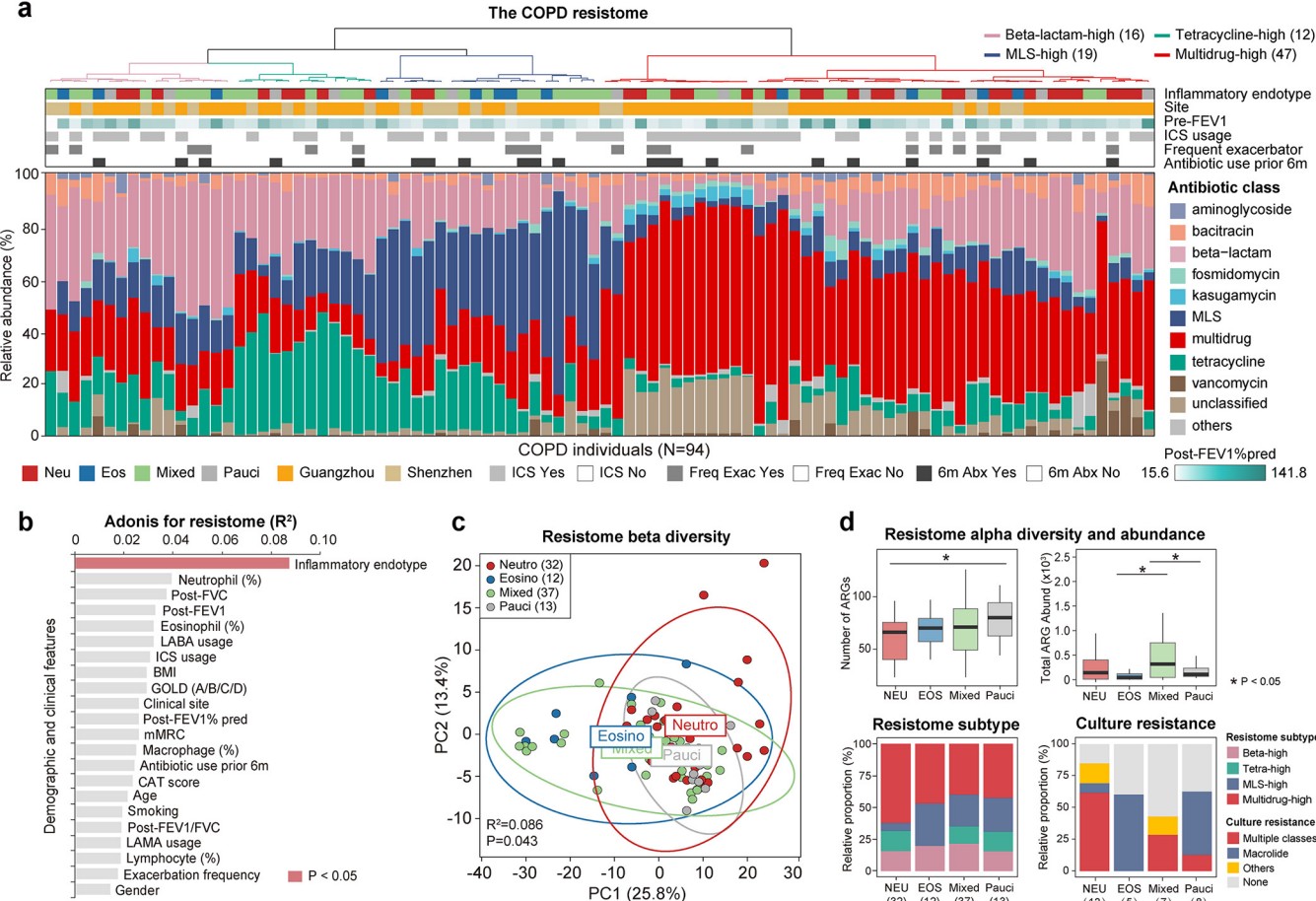

**FIG 1** The inflammatory endotype is associated with the airway resistome in COPD. (a) Unbiased clustering was performed on the ARG profile of all COPD participants using Ward's method, which revealed four resistome subtypes. The ARG profile, inflammatory endotype, site, postbronchodilator FEV1, ICS usage, antibiotic usage in the preceding 6-month period (6m Abx), and exacerbation frequency (Freq Exac) for each participant are shown. (b) Permutational multivariate analysis of variance using Adonis shows the association between the resistome and participant demographic and clinical features. (c) Principal-component analysis on the resistomes of all participants, colored by the inflammatory endotypes. (d) Box plots show the diversity and total abundance of ARGs in participants with different inflammatory endotypes. Histograms show the distribution of resistome subtypes and the resistance pattern in antimicrobial susceptibility test in participants with different inflammatory endotypes. Neu, neutrophils; Eos, eosinophils; Mixed, mixed granulocytes; Pauci, pauci granulocytes; pred, prediction; Abund, abundance; FEV1, forced expiratory volume in the first second; ICS, inhaled corticosteroids; FVC, forced vital capacity; LABA, long-acting beta-agonists; BMI, body mass index; GOLD, global initiative for chronic obstructive lung disease; mMRC, modified medical research council; CAT, COPD assessment test; LAMA, long-acting muscarinic antagonist.

multidrug-high resistomes in Guangzhou participants than in Shenzhen participants, the high representation of multidrug-high resistomes in neutrophilic COPD patients and MLS-high resistomes in eosinophilic COPD patients were observed in both sites (Fig. S2). For 33 participants with extra sputum available, their sputum was subjected to bacterial culture, matrix-assisted laser desorption ionization–time of flight mass spectrometry (MALDI-TOF MS) identification, and antimicrobial susceptibility testing in the clinical laboratory. In 8 of 13 neutrophilic participants, the bacteria cultured from their sputum exhibited resistance to multiple antibiotic classes, while bacterial isolates from 3 of 5 eosinophilic participants showed specific resistance to macrolides (Fig. 1d and Table S3). The results were partially in agreement with an endotype-related antimicrobial resistance pattern, although they should be interpreted with caution given the small sample size.

Examining the association of individual ARGs with sputum neutrophil and eosinophil percentages further revealed an endotype-specific correlation pattern. The abundance of 22 and 20 ARGs was correlated with sputum neutrophil and eosinophil percentages, respectively, whereas no ARGs were correlated with both (false discovery rate [FDR]-adjusted $P$ value of <0.05; Fig. 2a). Neutrophil-associated ARGs were mostly

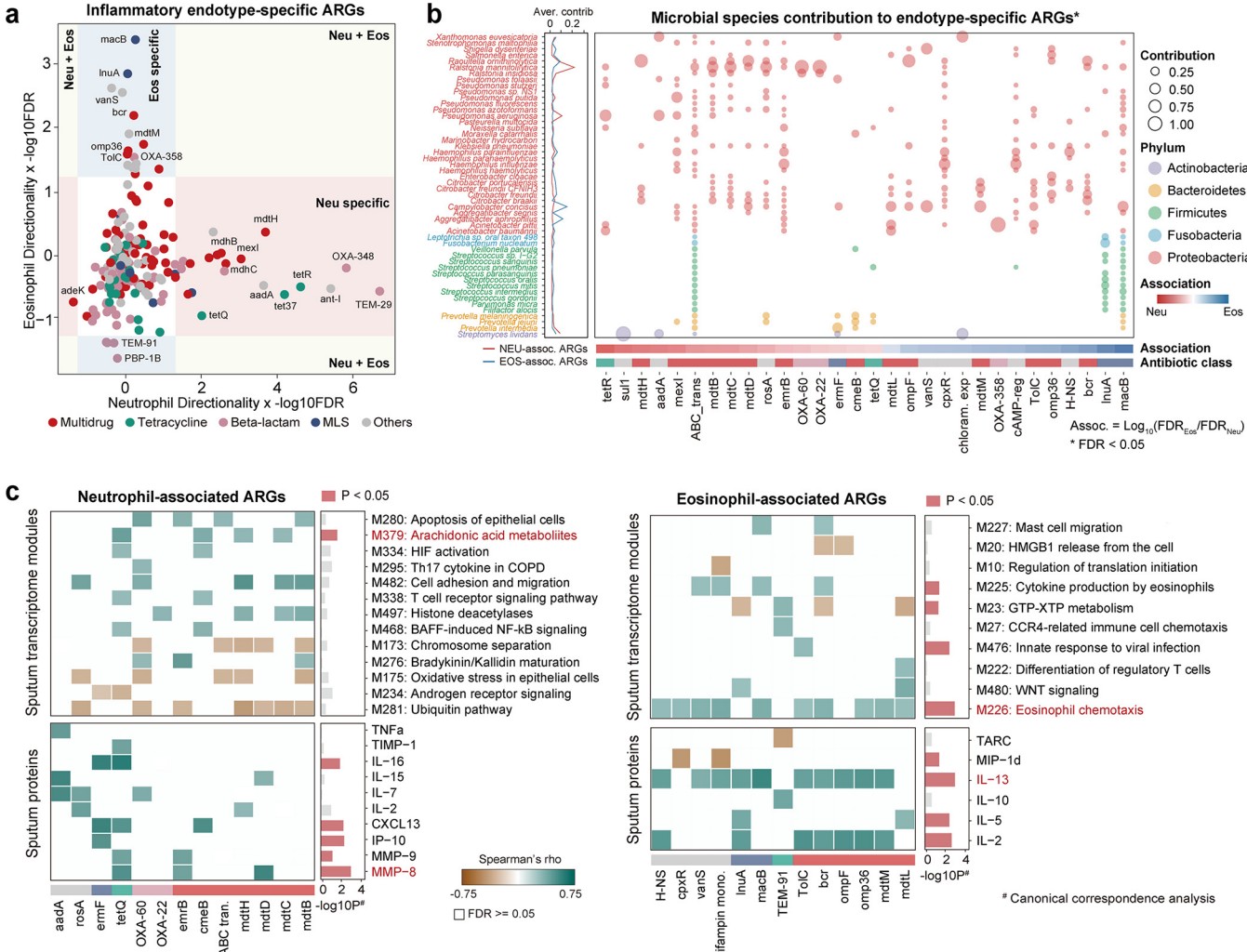

**FIG 2** The association between antibiotic resistance genes (ARGs) and inflammatory endotype, host transcriptome, and inflammatory mediators in COPD. (a) The association between ARGs and sputum neutrophil and eosinophil percentages. Each dot represents an ARG colored by antibiotic class. The $x$ and $y$ axes represent the directionality of correlation multiplied by the $-\log_{10}$ of the FDR-adjusted $P$ value. The ordination space is divided into colored area for association with neutrophils only, eosinophils only, both, and neither (FDR $P < 0.05$). (b) Bubble chart illustrating the relative contribution of bacterial species to the ARGs significantly correlated with sputum neutrophils or eosinophils. The average contribution of each species to the neutrophil- or eosinophil-associated ARGs is indicated beside the chart. (c) Heat maps illustrating the correlation of neutrophil- and eosinophil-associated ARGs with sputum host transcriptome modules and inflammatory mediators. The $-\log_{10} P$ values for the associations of modules and mediators with the resistome profiles in the canonical correspondence analysis for participants of neutrophilic or eosinophilic endotypes are shown beside the heat maps. The top module and mediator in the association are highlighted in red. Aver. contrib, average contribution; assoc, association.

of multidrug resistance, including *mdtBCDHL*, *emrB*, and *mexI* (Table S4). Eosinophil-associated ARGs were related to a diverse range of antibiotic classes, among which the *macB* and *lnuA* genes of MLS resistance were most significant (Table S4). *Ralstonia* spp. and *Pseudomonas* spp. were the top contributors to the neutrophil-associated ARGs, whereas species in *Campylobacter* and *Aggregatibacter*, previously found to be associated with sputum eosinophilia (13), contributed most to the eosinophil-associated ARGs (Fig. 2b).

These findings led us to further test the association between ARGs and host gene expression and inflammatory mediators. We obtained host transcriptomes from induced sputum for all participants by mRNA sequencing using an Illumina NovaSeq. A panel of sputum inflammatory mediators was characterized for 43 participants, with extra sputum samples available using antibody microarray, as described previously (RayBiotech, Norcross, USA) (14). Genes in the transcriptome were subject to coabundance clustering using a weighted correlation network analysis and pathway enrichment analysis (15) to obtain coexpression modules with functional annotation. For

modules significantly enriched for marker genes for macrophages, neutrophils, and eosinophils, (16), their expression levels were highly correlated with sputum differential cell count of the corresponding cell type (Fig. S3), suggesting the ability of the host transcriptome in sputum to reflect patient immune and inflammatory status. Thirteen modules and 10 sputum mediators were significantly correlated with the neutrophil-associated ARGs, among which M379 enriched with arachidonic acid metabolism and matrix metallopeptidase 8 (MMP-8) exhibited the strongest associations with overall resistome profiles in neutrophilic-predominant COPD participants (canonical correspondence analysis, $P < 0.05$; Fig. 2c and Table S5). Ten modules and 6 mediators were significantly correlated with eosinophil-associated ARGs, with M226 enriched with eosinophil chemotaxis, and IL-13 showing greatest associations with the resistome in eosinophilic-predominant participants (Fig. 2c and Table S5).

Taken together, these results uncover a previously unrecognized link between inflammatory endotype and the airway resistome in COPD. For neutrophilic-predominant COPD, the resistome was dominated by multidrug resistance genes from respiratory pathogens. For eosinophilic-predominant COPD, the resistome was diverse, with an increased portion of patients showing an MLS-high resistome. Our results suggest that the varied resistome across inflammatory endotypes could be a result of the underlying differences in the resident microbiota, whereas prior antibiotic history may be of less relevance, based on current data and literature (7, 8). These findings have potential clinical implications. A multidrug-high resistome driven by pathogenic taxa in neutrophilic COPD patients at the stable state may be a risk factor for antibiotic treatment failure during an exacerbation. Eosinophilic COPD is potentially associated with less frequent bacterial infection (17), but it nevertheless harbors a diverse range of ARGs, with a greater abundance of those being macrolide resistant. Macrolides are increasingly used as prophylactic antibiotics for the management of frequent exacerbation in COPD (18). The ARGs of macrolide resistance can be of concern, as macrolides may be used for eosinophil-high COPD patients who exacerbate despite inhaled corticosteroid-containing therapy. Multiomic analysis indicates that differential host pathways and sputum mediators (MMP-8 and IL-13) are associated with the resistome in patients with neutrophilic or eosinophilic inflammation, suggesting endotype-specific host-resistome interactions. These results raise intriguing possibilities that merit further investigation, such as whether these mediators could be markers that inform antibiotic selection and whether modification of host pathways may alter airway microbiota and antibiotic resistance.

The limitations of this study include its relatively small number of participants and cross-sectional nature. Despite having a validation cohort, the generalizability of the findings to patients with different demographic backgrounds and antibiotic history needs to be further assessed. In addition, it is important to assess the baseline resistome in healthy individuals in comparison with COPD patients. A previous study identified both shared and unique ARGs in healthy individuals and patients across a broad range of respiratory diseases, suggesting a complex airway resistome with both a core and discriminatory set of elements (7). Our results highlight the potential need for considering patient inflammatory endotype in decision-making about antibiotic use in COPD and the broader chronic respiratory diseases. Longitudinal surveys on the dynamics of the COPD airway resistome and its association with antibiotic treatment outcome are warranted to test the clinical applicability of resistome assessment in antibiotic management in COPD.

**Data availability.** The raw multiomic sequencing data have been deposited in the Chinese National GenBank Nucleotide Sequence Archive (CNP0001954). The computer code has been deposited at https://github.com/wangzlab/COPD_Resistome.

## SUPPLEMENTAL MATERIAL

Supplemental material is available online only.
**SUPPLEMENTAL FILE 1**, PDF file, 1.1 MB.

## ACKNOWLEDGMENTS

This work is supported by the National Key R&D Program of China (2017YFC1310600), the National Natural Science Foundation of China (31970112, 32170109, and 41907211),

the Science and Technology Foundation of Guangdong Province (2019A1515011395), and the Shenzhen Science Technology and Innovative Commission (KCXFZ202002011008256).

X.Y., Y.L., H.L., X.L., J.Y., J.G., Y.Y., Z.L., F.W., D.C., L.W., W.S., R.C., and Z.W. have nothing to declare. M.R.S. is an employee of CSL Behring. P.W.J. is an employee of GlaxoSmithKline.

Z.W. conceived the study design. H.L., Y.Y., Z.L., F.W., D.C., L.W., and W.S. contributed to clinical sample and data collection. X.Y., Y.L., H.L., X.L., J.Y., and J.G. performed statistical and bioinformatic analyses. M.R.S. and P.W.J. assisted in data analyses and interpretation. R.C. and Z.W. cosupervised the study. Z.W. drafted the manuscript. All authors provided critical revisions and approved the final manuscript.

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
