## [Reviewer comments · Microbiology Spectrum]

Microbiology Spectrum

Inflammatory Endotype Associated Airway Resistome in Chronic Obstructive Pulmonary Disease

Xinzhu Yi, Yanjun Li, Haiyue Liu, Xiaomin Liu, Junhao Yang, Jingyuan Gao, Yuqiong Yang, Zhenyu Liang, Fengyan Wang, Dandan Chen, Lingwei Wang, Weijuan Shi, David Lam, Martin Stampfli, Paul Jones, Rongchang Chen, and Zhang Wang

Corresponding Author(s): Zhang Wang, South China Normal University

Review Timeline:

Submission Date:	December 11, 2021
Editorial Decision:	January 8, 2022
Revision Received:	February 16, 2022
Accepted:	March 2, 2022

Editor: Xiaoyu Tang

Reviewer(s): Disclosure of reviewer identity is with reference to reviewer comments included in decision letter(s). The following individuals involved in review of your submission have agreed to reveal their identity: Jianmin Chai (Reviewer #3)

Transaction Report:

DOI: <https://doi.org/10.1128/spectrum.02593-21>

January 8, 2022

Dr. Zhang Wang
South China Normal University
Guangzhou
China

Re: Spectrum02593-21 (Inflammatory Endotype Associated Airway Resistome in Chronic Obstructive Pulmonary Disease)

Dear Dr. Zhang Wang:

Link Not Available

Sincerely,

Xiaoyu Tang

Journals Department
Reviewer comments:

Reviewer #1 (Comments for the Author):

This work by Yi et al performed in-depth investigation of respiratory microbiome, in particular its resistomes. Since a number of respiratory diseases are linked to microbiome differences, and that treatment and resistomes are tightly interacting with each other, their finer scale profiling of resistome with regard to endotypes of COPD is of potential interest and clinical relevance.

The authors overall did a good job in metagenomic analysis and ARG profiling, and found intriguing linkage between endotypes and resistomes. I suggest some more details to be refined to reflect the intrinsic questions remaining to be addressed or discussed in this manuscript.

1) the analysis of sputum transcriptome: is this widely applied in COPD (or other respiratory diseases)? How technically reliable is the host gene transcription in sputum and does it reflect well the immune status?

2) chicken and eggs: are endotypes the cause of resistome differences (e.g as a result of medicine usage differences in endotypes?) or the result of microbiome differences (what are the untreated and treated COPD microbiome like?), i guess medicinal history is important to address this question.

3) out of scope but important to discuss: baseline in healthy individuals, how much comparison can be performed?

Reviewer #2 (Comments for the Author):

The authors provide a very interesting report on COPD. The manuscript is well organized but some issues need to be addressed before it is ready for publication.

Line 110: There is no description or ticks on the Y-axis in the bar chart in Figure 1. It should be corrected.

Line 111: Procrustes analysis was used here to show the association between resistance and microbial composition, but there are no figures or tables to support this conclusion. The authors should provide further result demonstration, figures or tables.

Line 112-114: Clearly, the authors here found that the association between resistance and demographic and clinical features was not as strong as described. In the PERMANOVA results, only the inflammatory endotype was significant, and it can be observed from Figure 1b that this significance may only be a statistically weak association. The authors need to provide further explanations here.

Line 117-120: The conclusions in Figure 1d and Figure S1 do not fully conform to the author's description: the results observed in the two hospitals are not consistent.

Line 123-128: Among the participants of the four subtypes, the high Multidrug-high subtypes were the most dominant. However, bacterial isolates from the five eosinophilic participants did not have any multiple drug resistance and all of them were macrolides resistance, so the results could not be mutually verified or even contradictory.

Line 134: It is not appropriate to use $P < 0.1$ as the criterion of significance. Actually, $P < 0.1$ should not be used as the criterion for all other test, which will make the analysis results questionable. Commonly we use $P < 0.01$ or $P < 0.05$.

Reviewer #3 (Comments for the Author):

This is well written paper which has a refined description and discussion. The results, organization and writing are all good. But after reading this paper, a few remarks and questions arise that require clarification. Generally, please give transition words or sentence. The whole manuscript is too refined, which is good. However, adding transition words or sentence could help readers understand this paper.

L101-103: If all participants had not long-term antibiotic use, why remove 5 participants from analysis.

L112-114. Pls add the figures of Procrustes analysis; add the corresponding contents into materials and methods.

L123 Sputum from 33 participants was subject to bacterial culture. Is there a standard to select these 33 participants? If so, please add. In addition, how many participants from each endotype were selected?

L146 is 'sputum transcriptome' same with the terms of 'host transcriptome'?

Pls add more content of discussion if possible.

Supplementary document

Four reagent controls were included for sequencing. All of controls are negative controls? Pls give more details.

Material: the author measure Sputum Inflammatory Mediators from a subset of patients. So, how many patients were selected? Why did not measure all samples.

Staff Comments:

Preparing Revision Guidelines

For complete guidelines on revision requirements, please see the journal Submission and Review Process requirements at

<https://journals.asm.org/journal/Spectrum/submission-review-process>. **Submissions of a paper that does not conform to Microbiology Spectrum guidelines will delay acceptance of your manuscript. "**

Please return the manuscript within 60 days; if you cannot complete the modification within this time period, please contact me. If you do not wish to modify the manuscript and prefer to submit it to another journal, please notify me of your decision immediately so that the manuscript may be formally withdrawn from consideration by Microbiology Spectrum.

Response to Reviewer Comments:

Reviewer #1 (Comments for the Author):

General comment: This work by Yi et al performed in-depth investigation of respiratory microbiome, in particular its resistomes. Since a number of respiratory diseases are linked to microbiome differences, and that treatment and resistomes are tightly interacting with each other, their finer scale profiling of resistome with regard to endotypes of COPD is of potential interest and clinical relevance.

The authors overall did a good job in metagenomic analysis and ARG profiling, and found intriguing linkage between endotypes and resistomes. I suggest some more details to be refined to reflect the intrinsic questions remaining to be addressed or discussed in this manuscript.

Response to general comment: We thank the reviewer for the positive remarks along with the very helpful suggestions for us to improve the manuscript. In response, we have performed an additional analysis to demonstrate the capability of sputum transcriptome in reflecting the host immune and inflammatory status. We have also provided additional contexts to discuss the possible cause and effect relationship between the airway resistome and the inflammatory endotype, and the baseline resistome in healthy individuals in comparison with COPD patients. Below please find our point-by-point response to the reviewer's comments.

Comment 1: the analysis of sputum transcriptome: is this widely applied in COPD (or other respiratory diseases)? How technically reliable is the host gene transcription in sputum and does it reflect well the immune status?

Response 1: We thank the reviewer for this comment. Sputum induction has been the gold standard clinical tool for assessing airway inflammation for chronic airway diseases(1, 2). Quality controlled, induced sputum has been widely used to understand the airway gene expression for patients with COPD(3, 4) and asthma(5-7). Sputum gene signatures have been applied to identify patients with acute exacerbations(7), different inflammatory phenotypes(6) and responsiveness to treatment(8). With respect to whether sputum transcriptome reflects host immune and inflammatory status, in a recent study by Peters et al.(5), the authors used sputum transcriptome analysis to define inflammatory patterns (T2-high and T2-low) for asthmatic patients. By using a co-abundance clustering followed by a gene set enrichment analysis, they identified sets of genes enriched for specific immune cell types and showed that the expression levels of these genes were highly correlated with the actually measured sputum immune cell counts, suggesting technical reliability of sputum transcriptome and its capacity in reflecting host inflammatory status.

To demonstrate the same capacity of our transcriptomic data, we performed an

analysis similar to Peters et al.(5). Weighted correlation network analysis (WGCNA) on the sputum transcriptomic profile resulted in 497 co-abundance modules, in which the modules of M439, M198 and M226 was most significantly enriched to marker genes for macrophage, neutrophil and eosinophil, respectively (**Author Response Figure 1**). Similar as the observation in Peters et al.(5), the expression level of these modules were highly correlated with the sputum differential cell count for the corresponding immune cell types in our study, confirming the capability of our sputum transcriptomic data in recapitulating patient inflammatory status and phenotypes.

These data are added to **Figure S3** in the revised manuscript and summarized in the main text which reads (Line 156): “For modules significantly enriched for the marker genes for macrophage, neutrophil and eosinophil(16), their expression levels were highly correlated with sputum differential cell count of the corresponding cell type (Figure S3), suggesting the capability of host transcriptome in sputum in reflecting patient immune and inflammatory status”.

Author Response Figure 1 (Manuscript Figure S3). Enrichment analysis for immune cell marker genes using sputum transcriptomic data. a) The enrichment of transcriptomic co-abundance modules to marker genes for specific cell types, according to Peters et al.(5). The minus log₁₀ P-value was shown for the enrichment of each module in each cell type. For modules with P<0.01, they were colored according to the color codes of the modules in WGCNA. **b)** For modules most enriched for the marker genes for macrophage, neutrophil and eosinophil, their expression levels were highly correlated with sputum differential cell count of the corresponding cell type.

Comment 2: chicken and eggs: are endotypes the cause of resistome differences (e.g as a result of medicine usage differences in endotypes?) or the result of microbiome differences (what are the untreated and treated COPD microbiome like?), i guess medicinal history is important to address this question.

Response 2: We thank the reviewer for this insightful comment. We agree with the

reviewer that it is important to understand the cause and effect relationship between the inflammatory endotype and the airway resistome. Although the current study was not designed to tackle causality, two lines of evidence implicate that the observed association between the inflammatory endotype and the resistome could possibly be a result of the microbiome differences, and prior antibiotic history may have limited impact on the endotype-resistome association, based on current data and literature.

- 1) Procrustes analysis revealed a significant correlation between the airway resistome and microbiome taxonomic composition in our study ($M^2=0.019$, $P<0.001$), suggesting the resistome was likely inherently shaped by the airway microbiota. In addition, the airway microbiota was reported to be significantly associated with COPD inflammatory endotypes both in previous studies(9) and in our cohort (Adonis $R^2=0.092$, $P=0.013$). Therefore, the varied airway microbiota that encode different sets of antibiotic resistance genes could possibly underlie the association between the resistome and inflammatory endotype.
- 2) All participants were free of antibiotics over the preceding one month. The prior antibiotic history over the last six months was not associated with the endotype (Fisher's exact test $P=0.386$) or the resistome (PERMANOVA $P=0.439$) in our study. Furthermore, no association was found between patient antibiotic history and the airway resistome in a previous metagenomic study across a broader range of chronic respiratory disorders including COPD, asthma and bronchiectasis(10), as well as in another study on COPD resistome using qPCR assays(11). Therefore, based on current data and literature, prior antibiotic history may have limited association with the airway resistome.

Together, these evidence suggests that the observed endotype-resistome association was more likely a result of the underlying microbiota differences, whereas antibiotic history seems to be of less relevance. Whether the inflammatory endotype is a cause or a consequence of the microbiota differences needs to be studied further. Nevertheless, we agree with the reviewer that the medical history remains an important factor impacting the resistome, and its effects warrant further investigation in larger cohorts preferably with different demographic backgrounds and antibiotic history, which we have discussed in the original manuscript (Line 194).

We added below statement to the revised manuscript (Line 173): *“Our results suggest that the varied resistome across inflammatory endotypes could be a result of the underlying differences in the resident microbiota, whereas prior antibiotic history may be of less relevance, based on current data and literature”*.

Comment 3: out of scope but important to discuss: baseline in healthy individuals, how much comparison can be performed?

Response 3: We thank the reviewer for this suggestion. We agree with the reviewer

that, although out of scope for the current study, it is important to assess the baseline resistome in healthy individuals and its difference with that in COPD patients. Mac Aogain et al. profiled the airway resistome for 13 healthy individuals and 41 patients with a broad range of chronic respiratory disorders (COPD, asthma, bronchiectasis). Their results revealed a complex airway resistome with the presence of both a core and discriminatory set of ARGs across health and disease status.

We added a few lines of discussion in reference to this study, which reads (Line 196): *“In addition, it is important to assess the baseline resistome in healthy individuals in comparison with COPD patients. A previous study identified both shared and unique ARGs in healthy individuals and patients across a broad range of chronic respiratory disorders, suggesting a complex airway resistome with both a core and discriminatory set of elements”*.

Reviewer #2 (Comments for the Author):

General comment: The authors provide a very interesting report on COPD. The manuscript is well organized but some issues need to be addressed before it is ready for publication.

Response to general comment: We greatly appreciate the positive remarks from the reviewer along with very helpful suggestions to improve the manuscript. In response, we have added the figure and description for the Procrustes analysis, updated all results using the $P < 0.05$ cutoff, and provided additional clarifications to more properly reflect the statistical associations, the clinical site differences, and differences between the resistome and culture results. Below please find a point-by-point response to the reviewer’s comments.

Comment 1: Line 110: There is no description or ticks on the Y-axis in the bar chart in Figure 1. It should be corrected.

Response 1: We thank the reviewer for pointing this out. We have now added the description (relative abundance (%)) and ticks to the Y-axis in **Figure 1**.

Comment 2: Line 111: Procrustes analysis was used here to show the association between resistance and microbial composition, but there are no figures or tables to support this conclusion. The authors should provide further result demonstration, figures or tables.

Response 2: At the reviewer’s suggestion, we now presented the figure for the Procrustes analysis between resistome and microbiome taxonomic composition as **Figure S1** in the revised manuscript (**Author Response Figure 2**). The Procrustes analysis was performed based on the coordinate matrices generated by PCoA using Bray-Curtis dissimilarity indices on both datasets, using the *procrustes* function in R

vegan package. We have revised the statement in the manuscript which reads (Line 112): “A significant association was found between the resistome and microbiota composition using Procrustes analysis based on Bray-Curtis dissimilarity indices ($M^2=0.19$, $P<0.001$, Figure S1)”.

We have also added a statement of methods for Procrustes analysis to the Methods section in the supplementary document which reads (Supplementary document, Line 109): “Procrustes analysis was performed to assess the correlation between the resistome and microbiota taxonomic composition, based on the coordinate matrices of both datasets generated by principal coordinate analysis using Bray-Curtis dissimilarity indices, using the procrustes function in R vegan package”.

Author Response Figure 2 (Manuscript Figure S1). Procrustes analysis based on Bray-Curtis dissimilarity matrices showed a significant correlation between the airway resistome and microbiome taxonomic composition.

Comment 3: Line 112-114: Clearly, the authors here found that the association between resistance and demographic and clinical features was not as strong as described. In the PERMANOVA results, only the inflammatory endotype was significant, and it can be observed from Figure 1b that this significance may only be a statistically weak association. The authors need to provide further explanations here.

Response 3: We agree with the reviewer that the association between the resistome and the inflammatory endotype was modest and at the borderline of significance ($P=0.043$). The relatively small sample size could be a factor for the overall lack of significance between the resistome and clinical features. To more properly reflect the statistics, we have revised our statement which reads:

Abstract (Line 38): “Among all demographic and clinical factors, patient inflammatory endotype was associated with the airway resistome”. (the word ‘strong’ removed)

Main text (Line 115): “Among all demographic and clinical features, the inflammatory endotype had a statistically significant, albeit modest association with the resistome (Adonis, $R^2=0.087$, $P=0.043$, Figure 1b-c, Table S2)”.

Comment 4: Line 117-120: The conclusions in Figure 1d and Figure S1 do not fully conform to the author's description: the results observed in the two hospitals are not consistent.

Response 4: We thank the reviewer for pointing this out. We agree with the reviewer that there were differences in the resistome profiles between the two clinical sites. Specifically, there was a greater proportion of multidrug-high resistome in Guangzhou versus Shenzhen participants. We have added a distribution barplot to the original Figure S1 (the new **Figure S2**) to reflect the site differences. Despite the site differences, the distribution pattern of the resistome subtypes in between different inflammatory endotypes (i.e. the high representation of multidrug-high resistome in neutrophilic COPD patients and MLS-high resistome in eosinophilic COPD patients) was generally similar between the two sites.

We have revised our statement which reads (Line 122): “Despite a relatively greater proportion of multidrug-high resistome in Guangzhou versus Shenzhen participants, the high representation of multidrug-high resistome in neutrophilic COPD patients and MLS-high resistome in eosinophilic COPD patients were observed in both sites (Figure S2)”.

Comment 5: Line 123-128: Among the participants of the four subtypes, the high Multidrug-high subtypes were the most dominant. However, bacterial isolates from the five eosinophilic participants did not have any multiple drug resistance and all of them were macrolides resistance, so the results could not be mutually verified or even contradictory.

Response 5: We thank the reviewer for pointing this out. For the five eosinophilic participants that had bacterial culture data, bacterial isolates from three of them were of macrolide resistance and the isolates from the other two participants did not show resistance to the tested panel of antibiotics. This pattern was indeed not fully congruent with the overall resistome pattern that multidrug-high resistome was the most highly abundant resistome subtype among all patients. However, it was partially in agreement with the finding regarding the greater representation of MLS (macrolide-lincosamide-streptogramin)-high resistome in eosinophilic COPD patients than the other patient subgroups, and together suggests that macrolide resistance was likely relevant for eosinophilic COPD patients. The small number of patients with culture data is clearly an important caveat in interpreting the results (n=5 for eosinophilic patients).

We have revised the statement which reads (Line 131): “The results were partially in

agreement with an endotype-related antimicrobial resistance pattern, although they should be interpreted with caution given the small sample size”.

We have deleted the statement “*These different patterns were largely consistent with the in vitro antimicrobial susceptibility testing results, suggesting genotype-phenotype translatability*” in the original manuscript, to avoid over-interpretation of the data.

Comment 6: Line 134: It is not appropriate to use $P < 0.1$ as the criterion of significance. Actually, $P < 0.1$ should not be used as the criterion for all other test, which will make the analysis results questionable. Commonly we use $P < 0.01$ or $P < 0.05$.

Response 6: We thank the reviewer for this suggestion. We agree with the reviewer that $P < 0.05$ should be used to as threshold for statistical significance according to standard practice. We have updated all results using $P < 0.05$ (or $FDR < 0.05$ when applicable) throughout the manuscript.

Reviewer #3 (Comments for the Author):

General comment: This is well written paper which has a refined description and discussion. The results, organization and writing are all good. But after reading this paper, a few remarks and questions arise that require clarification. Generally, please give transition words or sentence. The whole manuscript is too refined, which is good. However, adding transition words or sentence could help readers understand this paper.

Response to general comment: We thank the reviewer for the positive remarks. In response, we have added the figure and description for the Procrustes analysis, and provided additional clarifications regarding the definition of sputum transcriptome and reagent controls, and the rationale for the selection of sputum samples for bacterial culture and measurement of inflammatory mediators. Transition words and sentences were also added accordingly. Please find below a point-by-point response to the reviewer’s comments.

Comment 1: L101-103: If all participants had not long-term antibiotic use, why remove 5 participants from analysis.

Response 1: Among the total number of 99 surveyed COPD patients, five participants were excluded due to missing record of long-term antibiotic use for these participants. For the remaining 94 ($99 - 5 = 94$) participants, they were all free from long-term antibiotic exposure. This was now made clear in the revised manuscript which reads (Line 101): “*Five participants with missing record of long-term antibiotic use information were excluded from analyses. All remaining 94 participants were free of antibiotics over the preceding one month and none had long-term macrolide use*”.

Comment 2: L112-114. Pls add the figures of Procrustes analysis; add the corresponding contents into materials and methods.

Response 2: At the suggestion of this reviewer and the Reviewer 2, we have now presented the figure for the Procrustes analysis between resistome and microbiome taxonomic composition as **Figure S1** in the revised manuscript (**Author Response Figure 3**). The Procrustes analysis was performed based on the coordinate matrices generated by PCoA using Bray-Curtis dissimilarity indices on both datasets, using the *procrustes* function in R vegan package. We have revised the statement in the manuscript which reads (Line 112): “A significant association was found between the resistome and microbiota composition using Procrustes analysis based on Bray-Curtis dissimilarity indices ($M^2=0.19$, $P<0.001$, Figure S1)”.

We have also added a statement of methods for Procrustes analysis to the Methods section in the supplementary document which reads (Supplementary document, Line 109): “Procrustes analysis was performed to assess the correlation between the resistome and microbiota taxonomic composition, based on the coordinate matrices of both datasets generated by principal coordinate analysis using Bray-Curtis dissimilarity indices, using the *procrustes* function in R vegan package”.

Author Response Figure 3 (Manuscript Figure S1). Procrustes analysis based on Bray-Curtis dissimilarity matrices showed a significant correlation between the airway resistome and microbiome taxonomic composition.

Comment 3: L123 Sputum from 33 participants was subject to bacterial culture. Is there a standard to select these 33 participants? If so, please add. In addition, how many participants from each endotype were selected?

Response 3: The selection of the 33 participants for bacterial culture was based on sample availability. When an extra sputum was available after sample processing for

metagenomics and other omics characterization, it was used for bacterial culture, MALDI-TOF identification, and antimicrobial susceptibility testing in the clinical laboratory. We have revised the statement in the manuscript which reads (Line 126): “For 33 participants with extra sputum available, their sputum was subject to bacterial culture, MALDI-TOF identification, and antimicrobial susceptibility testing in the clinical laboratory”.

The number of participants from each endotype with culture data was: NEU=13, EOS=5, Mixed=7, Pauci=8. These numbers were indicated in **Figure 1d** and **Table S3**.

Comment 4: L146 is 'sputum transcriptome' same with the terms of 'host transcriptome'? Pls add more content of discussion if possible.

Response 4: The reviewer is correct that, in our study, 'sputum transcriptome' is the same as 'host transcriptome' as only human protein-coding genes were characterized using mRNA-Seq in our study. Sputum transcriptome generally refers to host transcriptome in previous studies(4-6), but by broader definition, we think that it can also include gene expression from microbiota (i.e. metatranscriptomics).

To avoid confusion, we have revised all statements where 'sputum transcriptome' was referred, so they now read:

(Line 149): “We obtained host transcriptome from induced sputum for all participants by mRNA-sequencing using Illumina NovaSeq”.

(Line 159): “...suggesting the capability of host transcriptome in sputum in reflecting patient inflammatory status”.

(Line 255): “...associated with sputum host transcriptome modules and inflammatory mediators”.

Supplementary document

Comment 5: Four reagent controls were included for sequencing. All of controls are negative controls? Pls give more details.

Response 5: The reviewer is correct that all the reagent controls were negative controls. They are the DNA extraction blanks in which nuclease-free water was used for genomic DNA extraction, library preparation and sequencing, performed side-by-side with the actual sputum samples. We revised the statement in the supplementary Methods which reads (Supplementary document, Line 47): “Four reagent negative controls were included for sequencing. These negative controls were DNA extraction blanks in which nuclease-free water was used for genomic DNA extraction, library preparation and sequencing, performed side-by-side with the

sputum samples".

Comment 6: Material: the author measure Sputum Inflammatory Mediators from a subset of patients. So, how many patients were selected? Why did not measure all samples.

Response 6: We thank the reviewer for pointing this out. The sputum inflammatory mediators were measured in a subset of 43 COPD patients. This was again due to sample availability as a large volume of sputum material was required for the simultaneous characterization of metagenomics, host transcriptomics and inflammatory mediators. We have further clarified this information in the revised manuscript which reads

Main text (Line 151): "A panel of sputum inflammatory mediators was characterized for 43 participants with extra sputum samples available using antibody microarray as described previously".

Supplementary document (Line 86): "...were measured in a subset of 43 patients with extra sputum samples available using custom antibody microarray".

References

1. Lacy P, Lee JL, Vethanayagam D. 2005. Sputum analysis in diagnosis and management of obstructive airway diseases. *Ther Clin Risk Manag* 1:169-79.
2. Dragonieri S, Tongoussouva O, Zanini A, Imperatori A, Spanevello A. 2009. Markers of airway inflammation in pulmonary diseases assessed by induced sputum. *Monaldi Arch Chest Dis* 71:119-26.
3. Singh D, Fox SM, Tal-Singer R, Bates S, Riley JH, Celli B. 2014. Altered gene expression in blood and sputum in COPD frequent exacerbators in the ECLIPSE cohort. *PLoS One* 9:e107381.
4. Wang Z, Maschera B, Lea S, Kolsum U, Michalovich D, Van Horn S, Traini C, Brown JR, Hessel EM, Singh D. 2019. Airway host-microbiome interactions in chronic obstructive pulmonary disease. *Respir Res* 20:113.
5. Peters MC, Ringel L, Dyjack N, Herrin R, Woodruff PG, Rios C, O'Connor B, Fahy JV, Seibold MA. 2019. A Transcriptomic Method to Determine Airway Immune Dysfunction in T2-High and T2-Low Asthma. *Am J Respir Crit Care Med* 199:465-477.
6. Baines KJ, Simpson JL, Wood LG, Scott RJ, Fibbens NL, Powell H, Cowan DC, Taylor DR, Cowan JO, Gibson PG. 2014. Sputum gene expression signature of 6 biomarkers discriminates asthma inflammatory phenotypes. *J Allergy Clin Immunol* 133:997-1007.
7. Fricker M, Gibson PG, Powell H, Simpson JL, Yang IA, Upham JW, Reynolds PN, Hodge S, James AL, Jenkins C, Peters MJ, Marks GB, Baraket M, Baines KJ. 2019. A sputum 6-gene signature predicts future exacerbations of poorly controlled asthma. *J Allergy Clin Immunol* 144:51-60 e11.
8. Berthon BS, Gibson PG, Wood LG, MacDonald-Wicks LK, Baines KJ. 2017. A sputum

gene expression signature predicts oral corticosteroid response in asthma. *Eur Respir J* 49.

9. Wang Z, Locantore N, Haldar K, Ramsheh MY, Beech AS, Ma W, Brown JR, Tal-Singer R, Barer MR, Bafadhel M, Donaldson GC, Wedzicha JA, Singh D, Wilkinson TMA, Miller BE, Brightling CE. 2021. Inflammatory Endotype-associated Airway Microbiome in Chronic Obstructive Pulmonary Disease Clinical Stability and Exacerbations: A Multicohort Longitudinal Analysis. *Am J Respir Crit Care Med* 203:1488-1502.
10. Mac Aogain M, Lau KJX, Cai Z, Kumar Narayana J, Purbojati RW, Drautz-Moses DI, Gaultier NE, Jaggi TK, Tiew PY, Ong TH, Siyue Koh M, Lim Yick Hou A, Abisheganaden JA, Tsaneva-Atanasova K, Schuster SC, Chotirmall SH. 2020. Metagenomics Reveals a Core Macrolide Resistome Related to Microbiota in Chronic Respiratory Disease. *Am J Respir Crit Care Med* 202:433-447.
11. Ramsheh MY, Haldar K, Bafadhel M, George L, Free RC, John C, Reeve NF, Ziegler-Heitbrock L, Gut I, Singh D, Mistry V, Tobin MD, Oggioni MR, Brightling C, Barer MR. 2020. Resistome analyses of sputum from COPD and healthy subjects reveals bacterial load-related prevalence of target genes. *Thorax* 75:8-16.

March 2, 2022

Dr. Zhang Wang
South China Normal University
Guangzhou
China

Re: Spectrum02593-21R1 (Inflammatory Endotype Associated Airway Resistome in Chronic Obstructive Pulmonary Disease)

Dear Dr. Zhang Wang:

Your manuscript has been accepted, and I am forwarding it to the ASM Journals Department for publication. You will be notified when your proofs are ready to be viewed.

Sincerely,

Xiaoyu Tang
Editor, Microbiology Spectrum
